# Comparison of Manual, Automatic, and Voice Control in Wheelchair Navigation Simulation in Virtual Environments: Performance Evaluation of User and Motion Sickness

**DOI:** 10.3390/s25020530

**Published:** 2025-01-17

**Authors:** Enrique Antonio Pedroza-Santiago, José Emilio Quiroz-Ibarra, Erik René Bojorges-Valdez, Miguel Ángel Padilla-Castañeda

**Affiliations:** 1Instituto de Investigación Aplicada y Tecnología (InIAT), Universidad Iberoamericana, Ciudad de México C.P. 01376, Mexico; enriquepedroza2012@gmail.com; 2Departamento de Estudios en Ingeniería para la Innovación (DEII), Universidad Iberoamericana, Ciudad de México C.P. 01376, Mexico; erik.bojorges@ibero.mx; 3Instituto de Ciencias Aplicadas y Tecnología (ICAT), Universidad Nacional Autónoma de México, Ciudad de México C.P. 04510, Mexico; miguel.padilla@icat.unam.mx

**Keywords:** wheelchair user interfaces, virtual reality, voice control, navigation performance, navigation systems, daily life scenarios, immersive environments

## Abstract

Mobility is essential for individuals with physical disabilities, and wheelchairs significantly enhance their quality of life. Recent advancements focus on developing sophisticated control systems for effective and efficient interaction. This study evaluates the usability and performance of three wheelchair control modes manual, automatic, and voice controlled using a virtual reality (VR) simulation tool. VR provides a controlled and repeatable environment to assess navigation performance and motion sickness across three scenarios: supermarket, museum, and city. Twenty participants completed nine tests each, resulting in 180 trials. Findings revealed significant differences in navigation efficiency, distance, and collision rates across control modes and scenarios. Automatic control consistently achieved faster navigation times and fewer collisions, particularly in the supermarket. Manual control offered precision but required greater user effort. Voice control, while intuitive, resulted in longer distances traveled and higher collision rates in complex scenarios like the city. Motion sickness levels varied across scenarios, with higher discomfort reported in the city during voice and automatic control. Participant feedback, gathered via a Likert scale questionnaire, highlighted the potential of VR simulation for evaluating user comfort and performance. This research underscores the advantages of VR-based testing for rapid prototyping and user-centered design, offering valuable insights into improving wheelchair control systems. Future work will explore adaptive algorithms to enhance usability and accessibility in real world applications.

## 1. Introduction

Wheelchair controls are devices designed to allow people with mobility disabilities to operate the wheelchair independently and comfortably [1,2]. These controls vary according to the individual needs of each person and can cover a wide range of options [3]. Several types of controls are used in wheelchairs, and selecting the appropriate control depends on the user’s abilities and preferences [4]. Currently, there is an interest in improving the type of wheelchair controls to provide greater comfort and safety to users, from joysticks to autonomous control schemes. The joystick is an intuitive control widely used in electric wheelchairs. It allows for controlling direction and speed through simple movements of the joystick in different directions [5]. In autonomous control, some wheelchairs are equipped with remote controls that enable users to operate their wheelchairs without directly touching the controls on the chair [5,6]. These controls are particularly useful for individuals with limited hand mobility. Similarly, other technologies, such as voice command controls, are being tested for implementation in these devices [7].

Although wheelchair interfaces have been developed and evaluated in controlled environments for more than two decades, simulators, particularly those based on virtual reality (VR), have emerged as valuable complementary tools for the assessment and training of these technologies. These platforms offer practical advantages, such as reducing the physical space required for testing, customizing scenarios to replicate users’ daily environments, and enabling repeatable testing under controlled conditions. Moreover, simulators provide a safe training environment and mitigate risks associated with initial physical trials, especially for inexperienced users or potentially hazardous settings.

Recent studies have demonstrated that virtual environments not only are effective for assessing wheelchair navigation skills but also have therapeutic applications, such as mobility skills training in simulated conditions that would be difficult or costly to recreate physically. For example, Levac et al. [8] highlight that VR simulators can enhance the transfer of navigation skills to physical environments by offering a controlled and safe setting for learning. Similarly, Charlton et al. [9] emphasize the use of virtual technology as an intervention for wheelchair skills training, underscoring its practical and therapeutic benefits in personalizing training and improving mobility skills.

Still, to ensure the safety of users, they are tested and evaluated first through simulations, especially in virtual reality environments [10,11]. This implementation enables individuals with physical disabilities to practice and improve their wheelchair-driving skills in a safe and controlled manner [12,13]. They can face different virtual challenges and scenarios, such as overcoming obstacles, navigating narrow spaces, or performing complex maneuvers. At the same time, users can receive real-time feedback on their performance and learn new movement strategies [14]. In addition to being a training tool, virtual reality offers therapeutic benefits [15]. Providing an immersive visual experience can help improve confidence, coordination, and physical rehabilitation. Users can strengthen their motor skills and work on their balance and stability.

Virtual reality environments consist of computer-generated scenes and objects that allow users to feel immersed in them [16]. The visualization of this environment is achieved through virtual reality headsets or similar devices. These simulations can recreate situations involving risk or requiring testing before implementation, such as medical procedures or the behavior of certain devices [17]. Another advantage is the possibility of connecting devices such as joysticks, motion-sensing gloves, microphones, or vests to achieve a more effective interaction between the user and the machine [18] in a safer and more economically simulated environment before its implementation in real environments. Virtual objects can be manipulated in various ways, such as using joysticks, voice commands, or video tracking, as seen in a study by Kinect [19]. Each alternative requires user learning, so the user’s experience is a key factor in choosing the control method [20]. A user can control a character using various devices, depending on the virtual scenario or the control method that suits them best. Manual controls are commonly used to manipulate physical and virtual devices due to their ease of use. This type of control has been used since the mid-20th century for military purposes [21] and has been adapted for other uses, such as entertainment in video games, robot control, and even motorized wheelchair operation due to its functional design [22]. Using voice commands has become popular as an alternative to operating video games. Users can interact with games more naturally and seamlessly by speaking directly to them, known as Natural Language Processing (NIP) [23]. In this work, a speech recognition tool is used [24]. Speech recognition is a technology that allows computers to understand and process the language and voice of individuals. Models transcribe speech into text or actions, allowing users to interact with technology through voice instead of typing or touch interfaces [25]. Currently, this option is being implemented in devices such as smartphones with voice assistants, tablets, smart speakers (Alexa, Siri, Google) [26], and even virtual games like “Scream Go Hero” [27], where the main character moves based on the player’s verbal commands. This type of control is advantageous as it allows users to interact with devices remotely without physical contact, although a quiet environment is required.

Within a virtual environment, some objects have programmed movements and make decisions as needed, meaning they change their behavior depending on the situation they encounter, which is known as machine learning [28]. This type of implementation can be observed in games such as “Pac-Man” [29] or “Ping Pong” [30], where enemies change direction as the character advances through the field. More complex systems can also be seen utilizing this machine learning discipline, such as autonomous vehicles, which are given a destination, plot a route, and avoid obstacles they may encounter [31]. Autonomous controls are primarily used in navigation and movement in both robots and commercial vehicles [32]. This control is constantly improving and can become a useful tool, especially for individuals with mobility issues. However, there are advantages and disadvantages to using these controls, which may be more comfortable or feasible for some individuals compared with others, depending on their operation or task. Manual controls, operated through a joystick, button controls, or other means that require user intervention, are easy to use and provide physical feedback, such as vibration, which produces a more natural sensation [33]. On the other hand, autonomous systems have the advantage of performing tasks and making decisions independently, taking various alternatives depending on the situation, making them convenient as they do not depend on user intervention [34]. However, autonomous systems can be challenging for individuals, especially those not used to technology. Additionally, repairing them in case of failure can be costly and complicated.

The adoption of autonomous navigation technology has been maturing into real-life applications, for example, in the automotive industry. This type of navigation can impact improving assistance devices for people with motor disabilities.

A relevant type of application is the incorporation of autonomous or semi-autonomous navigation schemes in wheelchairs for patients with motor disabilities. Wheelchair controls are devices designed to allow people with mobility disabilities to operate the wheelchair independently and comfortably. Ideally, these controls could be personally configured according to the individual needs of each person, and their behavior could be adapted to cover a wide range of options.

However, designing autonomous control strategies for wheelchairs presents important challenges. An alternative is establishing virtual simulation environments, which allow exhaustive experimentation on the effectiveness of various control strategies economically and without putting patients at risk.

Voice controls are reliable and fast, particularly when it comes to tasks related to text generation, and they have high accuracy [35]. They are used for tasks in intelligent assistants or remote operations in education, healthcare, security, military, or entertainment industries [36]. However, in a voice-interacting system, they need to be in quiet environments or have a high signal-to-noise ratio to accurately identify voice, which can be a disadvantage in crowded or noisy places [37]. Each control system has specific characteristics that facilitate task execution compared with others. For example, in a study by Astuti [38], voice commands were used to start the engine and control a car. Similarly, in autonomous control systems, obstacles can be identified along the way, as observed in a study by Chatziparaschis [39], where a rescue robot makes decisions instead of an expert, who would operate the system manually.

In another study, Jayakodi [40] implemented a voice command module in a physical wheelchair in a controlled environment, enabling smooth movement and quick response to user requests. On the other hand, Alim [41] incorporated voice commands through a microphone and an Arduino controller in a wheelchair, which responds to the user’s movement instructions until it detects an object in its path. Infrared sensors were installed around the wheelchair for this purpose. It is interesting to see how these controls are being implemented in various fields and applications, whether to improve the mobility of people with physical disabilities, operate vehicles, or explore and rescue in dangerous areas, among others. The use of voice recognition technologies and autonomous systems enables a higher degree of interaction and control with devices and systems, which can positively impact people’s quality of life and the efficiency and safety of various industries. It is important to note that although these controls offer many advantages, evaluating their safety and reliability is also necessary. Autonomous systems can have failures and errors, and appropriate safety measures must be implemented to prevent accidents and operational problems. Additionally, it is important that these controls are accessible and easy to use for all individuals and that exclusion or discrimination due to lack of access or technical knowledge is avoided.

The use of these control systems may be more convenient or practical for users, depending on the purpose or conditions of the individual. However, to ensure their safety, it is necessary to conduct tests in virtual environments before implementing them on a physical device.

Summing up, to evaluate the best possible behavior of a wheelchair control system and minimize risks, simulation presents itself as an adequate research method. For example, in a study by Faria [42], a virtual wheelchair controlled by a physical device (joystick) was implemented. In contrast, in a research carried out by Harkishan [43], an autonomous control system is shown, where the wheelchair follows a trajectory until reaching its destination.

With these motivations, this paper presents an experimental study comparing three types of driving control systems (manual, voice, and autonomous) implemented in a wheelchair in a virtual reality environment to determine their effectiveness, usability, and ease of use in everyday scenarios. These control systems were evaluated in three virtual scenarios designed to represent typical daily life environments, such as a supermarket, a museum, and a city. The structure of this document is divided into five sections: introduction (Section 1), methodology and materials used (Section 2), results obtained (Section 3), discussion of the results (Section 4), and conclusions (Section 5).

## 2. Materials and Methods

The evaluation of wheelchair interfaces has traditionally been conducted in controlled environments using able-bodied participants or mannequins to ensure consistency, safety, and effectiveness in performing desired mobility tasks. However, the use of virtual reality (VR) simulators represents a valuable and complementary tool during the early stages of development and validation.

VR simulators provide a safe environment for preliminary testing, eliminating risks associated with the physical use of interfaces during early design phases. Moreover, they enable the customization and replication of complex, hazardous, or costly scenarios that are challenging to recreate in physical settings, such as congested urban environments or adverse weather conditions. These virtual environments also facilitate the collection of detailed data on user performance, including movement patterns and response times, enhancing study replicability and optimization.

The results obtained in virtual environments, however, present certain limitations when translated to physical settings. Factors such as sensor accuracy, the interaction of the interface with existing wheelchair hardware, and physical environmental conditions (e.g., flooring type or lighting) can significantly influence system performance. Therefore, VR does not aim to replace physical testing but rather serves as a complementary tool that allows for initial adjustments to the interfaces and guides the development of more robust technologies before subjecting them to evaluations in real-world controlled environments.

The simulator consists of two functional modules. First, there is a physical workstation that serves as a starting point for users. Second, the simulation module based on virtual reality technology is responsible for recreating immersive scenarios. The physical workstation is equipped with specialized software that allows the recreation of these virtual environments. In addition to the virtual reality goggles used by users to visualize these environments, a screen is available to observe participants’ movements and provide assistance if necessary.

### 2.1. Workstation

Virtual environment testing was conducted in a specialized computer laboratory equipped with the following components: an HP 280 G3 computer running a Windows 10 operating system, an Intel Xeon processor, a 256 GB SSD storage capacity, and a 64 GB RAM. The computer was also used to collect user information during the experimental interactive sessions with the users.

### 2.2. Software and Virtual Tools

The Unity platform was chosen as the main development engine due to its easy integration with virtual environments and virtual reality headsets [44]. The virtual simulation and computer connection were implemented using the C# programming language [45]. For visualization of the virtual environment, Oculus Rift [46] virtual reality headsets with a 128 GB storage capacity were used, along with an additional screen that allowed the technical team to observe the application in operation and provide timely instructions to the user (see Figure 1).

### 2.3. Setup Integration

Once the virtual scenario was implemented, it was necessary to visualize it using the Oculus virtual reality headsets and an additional screen. The virtual reality headsets allowed for the observation of the scenario in a 360° environment, enabling the exploration of every designed corner. The extra screen allowed the team to observe the movements of the user wearing the virtual reality headsets. The connection between the Unity platform and the virtual reality headsets is illustrated in the diagram shown in Figure 2.

### 2.4. Implemented Controls

Three types of controls were implemented to control and navigate the virtual wheelchair: manual control, autonomous control, and voice command control.

#### 2.4.1. Manual Control

The joystick is one of the most commonly used controls to interact with objects in virtual environments. This control consists of a lever rotating around its axis, and its movement is projected onto the controlled object. In this case, the user can operate the wheelchair using this control, which is implemented in the controls of the Oculus Rift headset. Due to its rotational nature, it allows free movement in any direction within the virtual scenarios, simulating the movement of electric wheelchairs.

#### 2.4.2. Autonomous Control

Autonomous control implemented in wheelchairs has emerged as a significant topic in the field of robotics. This control refers to the implementation of systems capable of emulating driving and navigation, providing greater autonomy and safety in the use of wheelchairs [47]. In this study, the use of the A (A-star) algorithm for route planning and raycasting for obstacle detection has proven to be an effective combination to optimize autonomous navigation in three-dimensional environments.

The A* algorithm [48] is a graph search method widely used in various applications, such as video games and route planning systems, due to its ability to find the shortest path between two nodes. This algorithm employs a heuristic function that calculates the cost of a path from a node to the destination, adding it to the accumulated cost of the path already traveled. Because of this combination, the algorithm selects the node with the lowest total cost, allowing for fast and efficient route searches [49].

In the implementation of this autonomous control system, the Unity development environment was used to simulate a three-dimensional space that included a virtual wheelchair, obstacles, and navigable areas. The wheelchair was equipped with physical components such as a Rigidbody and a Collider, enabling interaction with the simulated environment. Route planning was performed through the implementation of the A* algorithm, which evaluates the nodes in the search space based on their accumulated and estimated costs, selecting the most promising node for expansion. This iterative process continues until the destination node is reached, at which point the path is reconstructed by following the parent nodes generated during the search.

To enhance safety and avoid collisions, the route planning system was complemented with an obstacle detection technique based on raycasting. This method involves projecting a ray from the wheelchair’s position in the direction of its movement, evaluating whether the ray intersects with any object in its trajectory. If a potential collision is detected, the system halts the wheelchair’s movement and recalculates the route, adding an extra layer of safety and adaptability to the navigation system.

During the conducted tests, the virtual wheelchair demonstrated the ability to navigate effectively in complex environments, finding optimal routes and avoiding obstacles in real time. The bounding circle technique [50] was used to define a safe area around the objects in the environment, allowing the wheelchair to maintain an appropriate distance from obstacles. This technique is particularly useful in dynamic environments where the sudden appearance of new obstacles may alter the initially planned route. The test results indicate that the combination of the A* algorithm and raycasting provides a robust and efficient solution for autonomous navigation. This application not only improves the autonomy and safety of users but also has the ability to adapt to dynamic environments and replan routes in real time, which can be applied to both virtual and real-world environments.

Regarding haptic feedback, during the development of the tests with autonomous control, no haptic device was available to provide physical feedback, such as vibrations, in the event of collisions. Initially, a visual response was considered, such as an on-screen message or a movement resembling a “shake” on the screen to indicate collisions. However, during the initial trials, it was observed that this option became bothersome for users due to the frequency of collisions and the constant presence of visual messages, which caused distraction.

As a solution, a simple sound, similar to a “beep”, was implemented, which was triggered every time a collision occurred. This approach allowed users to detect collisions without interfering with the overall interaction or causing distraction.

On the other hand, regarding background noise, environmental noise was deliberately not included in the trials, as the main focus was on evaluating voice control. Nevertheless, it is acknowledged that background noise could influence the accuracy of voice commands, and this aspect is planned to be evaluated in future research.

#### 2.4.3. Voice Control

Voice control or voice commands are natural instructions or communication between a user and a computer, where the computer understands and executes the requested instructions [51]. This technology offers significant advantages, such as the possibility of not needing hands or a screen, a generally simpler user interface, and the ability to control the computer remotely. In this case, the “Speech Recognition API” [52] was used, a package defined in Unity capable of recognizing voice in multiple languages. This package works through logical grammar models [52], which allows the understanding of the structure of a language, giving coherence to a series of words or sentences and understanding their meaning. To use this package, it was necessary to implement a dictionary and create a list of words linked to specific movement instructions. These words are listed in Table 1.

During the initial pilot tests with voice control, it was observed that some participants experienced difficulties remembering specific commands required to navigate to a particular point within the virtual environment. In several instances, users attempted to use words similar to the established commands or paused their interaction to ask for the correct command before proceeding. This behavior highlighted the need to design a more flexible and accessible system for users.

As a solution, a repertoire of commands with similar meanings, such as “move”, “forward”, and “advance”, was implemented, allowing the participants to use any of these options to achieve the same action. This strategy aimed to increase users’ confidence and fluency when interacting with the system, reducing potential frustrations related to command precision. Despite the benefits observed, it is acknowledged that this flexibility might impact the clarity of the system. This issue will be addressed in future work to improve the precision and consistency of the voice control interface.

### 2.5. Experimental Virtual Reality Scenarios

Three virtual scenarios of everyday use were employed to observe and compare the implemented controls. These scenarios included a supermarket, a museum, and a city, as explained below.

#### 2.5.1. Supermarket

This scenario represents an enclosed space of a typical supermarket scenario in a daily life situation. It was designed with 10 aisles, 14 shelves, 2 refrigerators, 5 checkout counters, 1 ATM, and a door that functions as an entrance and exit, as shown in the floor plan in Figure 3. Due to its narrow space, caution must be taken with surrounding objects to avoid collisions, as would happen in a real supermarket. The task to complete in this scenario is to collect five products distributed throughout the environment to gather data related to the different types of implemented control, avoiding random obstacles such as moving people or shopping carts. An example of a traversal through the scenario is shown in Figure 4.

#### 2.5.2. Museum

This scenario has wider corridors and more distant objects than the previous one. It has a total of 3 exhibition rooms—the Egyptian room, the Oriental room, and the Indian room—which are connected through an access, as can be seen in the floor plan in Figure 5. Each room contains a total of 4 exhibits in the Oriental room, 6 objects in the Egyptian room, and 8 objects in the Indian space. In this scenario, 5 objects related to the museum were distributed, which the users had to retrieve while avoiding obstacles.

#### 2.5.3. City

This scenario represents a part of a city with streets, buildings, shops, sidewalks, parks, and people, among other elements. It is classified as an open place with no clear edges or boundaries, as objects are connected through streets. The scenario features 22 buildings and 4 primary and 3 secondary streets (Figure 6). In this scenario, 5 objects are distributed throughout the space, but people move randomly and can obstruct the user’s path, requiring the user to choose an alternative route.

## 3. Experiments

### 3.1. Objective of the Study

The main objective of the present study was to evaluate the effectiveness of the controls used, as well as their ease of use in each implemented scenario and the acceptance by the users. Additionally, this study aimed to investigate how these tools can improve the user experience and navigation for wheelchairs. Furthermore, the influence of motion sickness on utilizing these controls was investigated, as well as the users’ adaptability to operating a wheelchair in a virtual or physical environment.

### 3.2. Participants

Twenty participants (9 females and 11 males) between the ages of 18 and 65 took part in this experiment. All participants were right-handed and did not have visual perception problems, mobility difficulties, or similar conditions. These criteria were chosen to ensure that the initial pilot study focused on assessing the system’s functionality and usability without external variables that could influence the outcomes. Future studies will expand the participant pool to include wheelchair users and individuals with relevant experience, providing expert insights into the system’s performance. Before participating, all volunteers provided written informed consent after being fully informed about the study procedures, objectives, and their right to withdraw at any time without repercussions.

This study was reviewed and approved by the Technical Research Council of the Graduate Program in Engineering Sciences at Universidad Iberoamericana, which assesses and evaluates the risks of research projects. The evaluation classified this study as minimal risk in accordance with Articles 100, 102, and 109 of the General Health Law on Research in Mexico. In compliance with the Declaration of Helsinki, all ethical principles were strictly followed, ensuring the safety, integrity, and well-being of the participants. For further details, please refer to the Ethical Statement section of this manuscript.

During the experiment, the participants were seated in a comfortable chair positioned 1 m away from a computer and a screen. They used virtual reality goggles to interact with the simulation, which ensured an immersive experience and minimized any potential discomfort.

### 3.3. Experimental Procedure

First, the participants were positioned in front of the computer wearing the virtual reality head-mounted display, an Oculus Quest 2 (Meta, 2020). Although the head-mounted display provided an immersive experience, the participants were seated 1 m away from a screen that displayed their movements and instructions during the simulation. This setup ensured a safe environment, allowing supervisors to monitor their performance and provide assistance if needed. The participants were seated in a comfortable chair to ensure stability and focus throughout the tests.

During the tests, the participants underwent prior training in an empty space with simple objects, such as geometric shapes, to familiarize themselves with controlling the wheelchair in a controlled environment. This training aimed to ensure that the participants were comfortable with the control system before proceeding to the main trials. During this phase, guidance was provided on how to operate the wheelchair and navigate through the space, allowing them to practice until they felt confident with the interaction.

Once the participants had mastered the basic use of the wheelchair, the evaluation trials were conducted randomly, without any indication of the target object locations. This randomization ensured that the trials reflected a more real-world scenario, where users must interact with objects without prior knowledge of their location.

Then, each participant completed three attempts in each scenario, for a total of nine attempts, using a different control in each one, with a 5 min rest between attempts, in a random counterbalanced order. A time limit of 5 min per scenario was established, although the participants could finish the tests in less time. The participants navigated each scenario in search of various target objects scattered throughout the place, picking them up and trying to avoid other obstacles to prevent collisions. Collisions occurred when the circumference assigned to the wheelchair encountered an object or wall in the scenario. The participants were instructed to adopt a comfortable posture, avoid sudden movements, or stand up during the session. They were informed that an assistant would always be available in case they needed to stop the game.

### 3.4. Measurements

Three ordinal-scale metrics (time, distance, and number of collisions) and one nominal-type metric (motion sickness) were measured from the simulation of the scenarios. These metrics helped analyze the participants’ movements and understand the relationship between their performance and the obtained data. To obtain the total time for each simulation, a stopwatch was used, which started when the wheelchair started moving and stopped when it reached the endpoint or exit. To measure the distance traveled, coordinates (x, y) were tracked from the origin to the final point where the simulation was stopped. Then the mean velocity and acceleration were calculated from the recorded path.

### 3.5. System Assessment

The questionnaire used to assess the experience consists of 25 questions designed to capture participants’ perceptions of various aspects of the navigation process. The questions address topics such as the level of fatigue experienced during navigation, the ease of controlling the wheelchair movements, and the degree of satisfaction with the wheelchair control system. These questions were scored using a Likert scale of 1 to 7, where 7 represents “strongly agree”, 6 “often agree”, 5 “sometimes agree”, 4 “neutral”, 3 “sometimes disagree”, 2 “often disagree”, and 1 “strongly disagree”. The Likert scale for the questions can be seen in Table 2.

Table 3 presents the complete statements of the questions administered after each trial, covering different scenarios and control combinations, with a total of nine questionnaires completed per participant.

### 3.6. Statistical Analysis

The system was evaluated in three different scenarios. The city scenario is characterized by being more open, without obstacles in the path, although it involves covering longer distances. On the other hand, the museum is a semi-enclosed space with objects scattered in the rooms. Lastly, the supermarket scenario stands out for having objects closer to each other and narrower aisles. All data were collected, and dispersion measures such as mean and standard deviation were calculated, as shown in Table 3. Subsequently, a series of multivariate analyses of variance (MANOVA) [53] following an experimental design of 3 Scenarios (supermarket, museum, city) X 3 Controls (manual, autonomous, voice) was conducted for the variables of Time, Distance, Velocity, Acceleration, and the number of Collisions. Bonferroni honestly post hoc tests [54] were subsequently applied per each MANOVA. Finally, a series of non-parametric Kruskal–Wallis tests [55] were applied to each perception questionnaire item, comparing the control navigation conditions. All analyses were performed using SPSS software version 27.

Although time and distance were used in this study as indicators of navigation performance, it is acknowledged that these metrics do not always directly reflect the ease of use of the interfaces. To address this, time and distance data were analyzed using MANOVA to determine significant differences between scenarios and control types. Additional correlation analyses were performed to explore the relationships between these metrics and questionnaire responses related to ease of use, comfort, and perceived control. For instance, it was verified whether longer navigation times correlated with lower scores on perceived ease of use.

Furthermore, these metrics were complemented with a Likert scale questionnaire, designed to assess the participants’ subjective perceptions. Results from the Kruskal–Wallis tests revealed significant differences in perceived ease of use, comfort, and control across different control types. Additionally, analyses related to physical and psychological comfort, such as the level of dizziness and discomfort reported during navigation, were included. These subjective assessments provided a more comprehensive understanding of user experience, highlighting factors beyond time and distance.

## 4. Results

### 4.1. Navigation Performance

The series of MANOVA revealed a main factor effect for the scenario for the five variables Time (F(2,177) = 547.33, *p* < 0.0001), Distance (F(2,177) = 5030.42, *p* < 0.0001), Velocity (F(2,177) = 13839.04, *p* < 0.0001), Acceleration (F(2,177) = 2546.53, *p* < 0.0001), and Collisions (F(2,177) = 319.76, *p* < 0.0001). Moreover, the consequent Bonferroni post hoc tests confirmed significant differences among the three scenarios for all the variables, indicating a different motion performance of participants during the navigation, depending on the virtual scenario. Table 4 displays the observed descriptive statistics for the five variables, compared by scenario. As can be observed, in general, the participants traveled longer paths in the city outdoor scenario with faster movements and fewer collisions than in the other indoor scenarios. On the other hand, they covered the shorter distance at a slower pace but collided more in the supermarket indoor scenario.

The same MANOVA tests performed also revealed a main factor effect for the type of control for most of the variables Time (F(2,177) = 587.31, *p* < 0.0001), Distance (F(2,177) = 357.166, *p* < 0.0001), Acceleration (F(2,177) = 371.14, *p* < 0.0001), and Collisions (F(2,177) = 1500.54, *p* < 0.0001), with exception of velocity. Again, the successive Bonferroni post hoc tests confirmed significant differences among the three control types for all the variables but for velocity, meaning a different motion performance of the participants depending on the type of control for the navigation. Table 5 shows the observed descriptive statistics for the five variables, compared by control type. In this case, the participants generally exhibited the best performance with autonomous control, completing the tasks in shorter periods and distances with remarkably much fewer collisions than the other controls. Contrarily, the poorest performance was exhibited when navigating under voice control regarding executing time and covered distance, which, together with manual control, resulted in much more collisions than in the automatic mode.

Finally, as shown in Table 6, the MANOVA tests also revealed statistical interaction among Scenario X Control combinations, meaning a different motion performance if navigating in different scenarios and depending on the navigation control scheme, but generally better using the autonomous navigation.

The boxplots in Figure 7 display a graphical comparison of the time taken for each scenario and control type. The boxplots in Figure 8 visually compare the distance covered for each scenario and control type. Finally, the boxplots in Figure 9 illustrate a comparison of the collisions that occurred for each scenario and control type.

### 4.2. Questionnaire Ratings and System Assessment

After conducting the perception questionnaire survey, two series of non-parametric Kruskal–Wallis tests were performed. These tests compared the types of controls and scenarios, evaluating differences in the perception of tiredness, ease, and pleasantness, separately for the wheelchair controls and the different scenarios.

The Kruskal–Wallis tests revealed significant differences among the types of controls for the three perceptual variables: tiredness (*p* < 0.013), easiness (*p* < 0.0001), and pleasantness (*p* < 0.031). Then Bonferroni tests revealed more specifically that a higher significant sensation of tiredness using manual control (mean score of 3.33 ± 1.42, median 4) than voice control (mean score of 2.93 ± 1.53, median 2) was perceived, as shown in Figure 10. For easiness, the participants perceived that is was significantly more difficult to use voice control (mean score of 3.23 ± 1.14, median 2) than manual control (mean score of 4.18 ± 0.62, median 5) and automatic control (mean score of 4.43 ± 0.67, median 5). For pleasantness, the perceived experience was positive for the three controls, with a significantly higher level for voice (mean score of 4.27 ± 0.71, median 5) than for manual control (mean score of 3.95 ± 0.56, median 5).

For the evaluation of the perceived differences among the different scenarios, the Kruskal–Wallis tests did not reveal significant differences between scenarios for the perceived sensation of tiredness and easiness, with a low level of tiredness (median score of 2) and a low level of difficulty (median easiness sensation of 4). The participants generally enjoyed navigating the three scenarios, but slightly significant differences were observed for pleasantness (*p* < 0.0001), to a lesser extent, within the supermarket (mean score of 3.77 ± 0.62, median 4) compared with that within the museum (mean score of 4.14 ± 0.63, median of 4) and the highest level in the city scenario (mean score of 4.41 ± 0.67, median 5).

Regarding the possible perceived side effects, it was observed first that both manual and autonomous control generated a higher frequency of dizziness while navigating with manual control, followed by autonomous control, with much less occurrence during voice control (Figure 11), in general for the three scenarios.

With respect to the discomfort perceived by the users (Figure 12), it can be observed that manual control generated a higher frequency compared with the other controls, especially in the museum scenario. Autonomous control was in second place, with the highest value also in the museum. Finally, the control that generated the least discomfort among users was voice control.

To ensure fairness and minimize potential habituation effects, the order of the scenarios was assigned randomly for each participant. Additionally, the participants were allowed to take a 5- to 10-min rest between scenarios if needed, which helped reduce fatigue and maintain optimal performance during the tests.

Thus, although manual and autonomous controls were considered easier to use by users, they were also associated with a higher sense of dizziness or discomfort. Conversely, voice command control, although more challenging to handle, generated less dizziness compared with the other two control types.

## 5. Discussion

This study highlights the potential of virtual reality environments as effective tools for designing and evaluating wheelchair navigation systems employing various control strategies. By analyzing user performance and perception across manual, autonomous, and voice-controlled methods, the findings provide valuable insights into the strengths and limitations of each approach. These results not only emphasize the importance of tailoring navigation systems to the specific needs of users but also underscore the challenges that must be addressed to enhance the accessibility and usability of these technologies in practical applications.

In this work, a virtual reality platform was presented to simulate navigation using wheelchairs, utilizing three types of controls: manual, automatic, and voice. The performance in navigation tasks was studied in a group of healthy participants, comparing the three types of controls in three typical daily life scenarios. Additionally, the adverse effects of dizziness and feelings of discomfort experienced by the participants during their immersive navigation experiences were investigated. The objective was to evaluate the potential of using autonomous and voice control in comparison with manual control, as well as to assess the potential of the proposed virtual environment as an experimentation tool in the design and use of wheelchairs.

Regarding performance, both autonomous and manual controls proved effective in maneuvering the wheelchair in the virtual environment. These methods allowed the participants to navigate the wheelchair precisely and quickly, suggesting a satisfactory response in terms of motion control. This can be attributed to the participants’ familiarity with manual controls and the careful implementation of autonomous control, which could adapt to users’ commands and requirements.

However, despite their better performance, autonomous and manual controls generated dizziness among the participants. This may result from the need to perform quick and precise movements in a virtual environment, which can overload the participants’ sensory and visual systems. Additionally, the lack of haptic feedback in autonomous control may have contributed to the dizziness, as the participants did not receive direct physical sensations when interacting with the virtual wheelchair.

On the other hand, voice control exhibited the poorest performance in terms of control accuracy and speed. The participants struggled to communicate the appropriate commands accurately and experienced delays in the wheelchair’s response. Although this control method did not generate as much dizziness as the others, its unsatisfactory performance suggests that voice recognition accuracy and capability improvements are necessary to make it viable in virtual environments.

The results obtained through statistical analysis indicate significant differences in average times among the scenarios, regardless of the type of control used. Specifically, it was observed that the average times in scenarios 1 (Supermarket) and 3 (City) are similar, while scenario 2 (Museum) has shorter times. This suggests that the scenario influenced the participants’ performance, with scenario 2 being more efficient regarding time, regardless of the type of control used. Additionally, significant differences were found in other variables, such as distance, speed, acceleration, and number of collisions among the control groups and scenarios.

The findings of this study demonstrate the potential of virtual reality (VR) simulators for the assessment and training of wheelchair skills. These results align with previous research highlighting the applications of VR technology in this domain. For instance, a work by Levac et al. [8] underscores the utility of virtual environments in wheelchair skills training, emphasizing their capacity to provide a controlled, safe, and repeatable setting, which aligns with the advantages identified in the present study.

Furthermore, Harrison et al. [56] analyzed the use of virtual reality for the assessment and training of novice powered wheelchair users. Their findings suggest that VR technology not only enhances the learning experience but also enables users to develop essential skills before facing real-world environments. Similar benefits were observed in this study, particularly regarding improved navigation skills and the mitigation of risks associated with training in physical environments.

A review conducted by Charlton et al. [57] on manual wheelchair training strategies highlights the importance of personalized methodologies to optimize user outcomes. In comparison, the VR-based approach presented in this study also offers a high degree of personalization, allowing virtual scenarios to be tailored to the specific needs of the participants. This adaptability could be particularly beneficial for inexperienced users or those in the early stages of training.

While the results of this study are promising, it is important to acknowledge the inherent limitations of VR simulators. As noted by Harrison et al. [56], factors such as the transferability of skills from virtual to physical environments and the accuracy of replicating real-world conditions remain significant challenges. Therefore, the findings of this research should be interpreted as complementary to evaluations conducted in traditional physical settings, as also suggested by Levac et al. [8] and Charlton et al. [9].

This study strengthens the existing literature by demonstrating that VR simulators can play a critical role as complementary tools for the assessment and training of wheelchair skills.

Regarding the use of voice control, our findings reveal similar limitations to those reported by Barriuso [58], who highlighted an increase in response time and errors when users relied solely on voice commands. However, the present study expands on these findings by demonstrating that the accuracy of voice control improves in simpler or open environments, such as a street or city, where navigational demands are reduced.

A study conducted by Lee [59] found a correlation between stress and motion sickness, in both simulated and real-world settings. The results of this research align, to some extent, with this finding, as a higher incidence of motion sickness was observed when users searched for an object or path against time pressure, compared with navigating a familiar route. Another factor contributing to motion sickness could be scenarios with constant visual stimuli, such as the supermarket, likely due to the increased sensory load.

Mu and He [60] observed that users quickly adapt to voice controls in virtual reality environments. The results of this study indicate that adaptation strongly depends on the type of environment. In more complex and dynamic settings, users showed greater difficulty adapting to voice commands, which could be related to the simultaneous need for visual and auditory processing in such contexts.

It is worth noting that these results are consistent with those from a work by Rebsamen [61], who mentioned that the context or environment in which a task is performed can significantly impact people’s performance. For example, a study by Yue [62] showed that familiarity with the environment, scenario complexity, or the presence of distractions can influence performance in navigation or driving tasks.

Overall, these findings highlight the importance of considering both performance and user perception when selecting a control method for wheelchairs in virtual environments. While autonomous and manual controls offer better performance, the dizziness associated with these methods must be addressed by reducing sensory and visual load. Voice control, although causing less dizziness, requires improvements in accuracy and recognition to be a viable option. These results provide a solid foundation for future research and improvements in designing and implementing wheelchair control systems in virtual environments.

Thus, although manual and autonomous controls were considered easier to use by users, they were also associated with a higher sense of dizziness or discomfort. Conversely, voice command control, although more challenging to handle, generated less dizziness compared with the other two control types.

Additionally, when evaluating the safety of the three control methods, autonomous control demonstrated superior performance, showing the lowest number of collisions across all scenarios. This result highlights the reliability of autonomous systems in minimizing errors during navigation. In contrast, manual control exhibited the highest number of collisions, reflecting the challenges users faced in maintaining precise navigation. Voice control, while generating fewer collisions than manual control, still showed a slightly higher number compared with the autonomous system.

The study conducted in this work provided helpful information about the efficiency and perception of controls in virtual environments simulating wheelchair navigation in different realistic scenarios. It aims to promote the implementation of experimental setups for assessing the effects of autonomous and voice command-based controls in virtual environments and to elucidate the possible configurations able to increase efficiency and reduce motion sickness in these environments. In general, these results can help improve the implementation of virtual reality in various fields and design better controls for a more effective and comfortable user experience. Thus, the experiences reported in this study may lead to the basis for novel studies in wheelchair navigation under different experimental conditions but in a risk-free and economical fashion.

However, it is important to consider the limitations of this study, such as the sample size and the generalization of the results to other contexts, and further research with a broader and more rigorous focus is recommended to confirm and expand these findings, such as having enhanced autonomous control and a better voice recognition algorithm. In addition, the experiences suggest that strategies should be considered to reduce sensory and visual load when implementing these control methods in virtual environments, such as providing haptic feedback to enhance perception and alleviate motion sickness.

## 6. Conclusions

This study highlights the valuable role of virtual reality environments in the design and evaluation of wheelchair navigation systems using various control strategies. The findings underscore the importance of balancing performance metrics and user perception when selecting a control method for virtual wheelchair navigation.

Experimental results indicate that autonomous and manual controls offer superior performance compared with voice control. However, the motion sickness associated with these methods must be addressed to ensure user comfort. Conversely, voice control, while causing less motion sickness, requires significant improvements in accuracy and recognition to become a viable alternative.

This research provides fundamental insights into how virtual environments can facilitate the development and evaluation of wheelchair navigation systems. It emphasizes the potential of virtual reality-based methodologies to inform the design of accessible and adaptable solutions for users with diverse needs.

This study contributes to the advancement of autonomous navigation technologies by offering a controlled platform to assess different control strategies. These insights serve as a foundation for future research aimed at optimizing and personalizing wheelchair navigation systems, thereby enhancing their accessibility and user experience.

## 7. Future Work

Future studies should focus on testing the system in a broader range of environments. While the virtual scenarios in this study replicated typical daily situations, real-world environments introduce additional variables, such as surface textures and lighting variations. These factors can significantly affect user experience and system performance, necessitating further optimization and validation.

Moreover, it is essential to incorporate haptic feedback during testing. This type of interaction can enhance user experience by providing physical cues that improve perception and control during navigation.

Another important avenue for future research involves evaluating the system with individuals who use wheelchairs, as they will directly benefit from these technologies. Conducting tests with real users will provide a more comprehensive understanding of the system’s usability and effectiveness under practical conditions.

Additionally, a comparison should be made between users with experience in virtual reality or video games and those without to examine how familiarity with these technologies influences motion sickness during tests. This comparison will help determine whether experienced users are less prone to motion sickness and adapt more easily to the system, while users without such experience may experience greater discomfort. These insights will be valuable for refining the system to ensure it accommodates both groups, offering improved accessibility.

This study represents an initial step in exploring autonomous navigation technologies for wheelchairs. It is crucial to continue this line of research in collaboration with healthcare professionals, rehabilitation experts, and end users. This approach will ensure that the system’s design aligns with the specific needs and expectations of the target population, enhancing its practical impact and clinical relevance. 

## Figures and Tables

**Figure 1 sensors-25-00530-f001:**
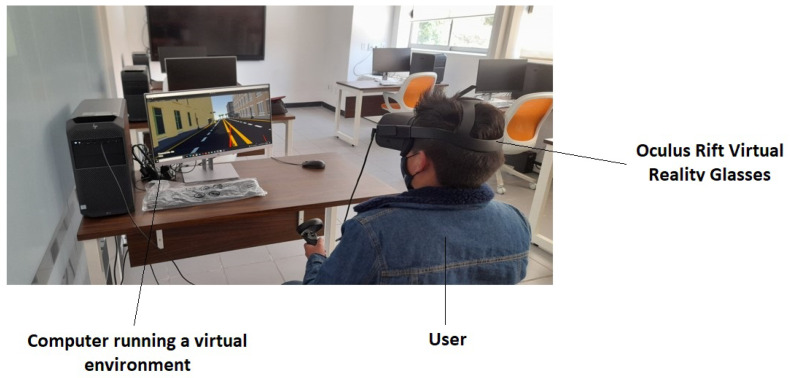
Experimental setup with a participant interacting with one of the virtual environments.

**Figure 2 sensors-25-00530-f002:**
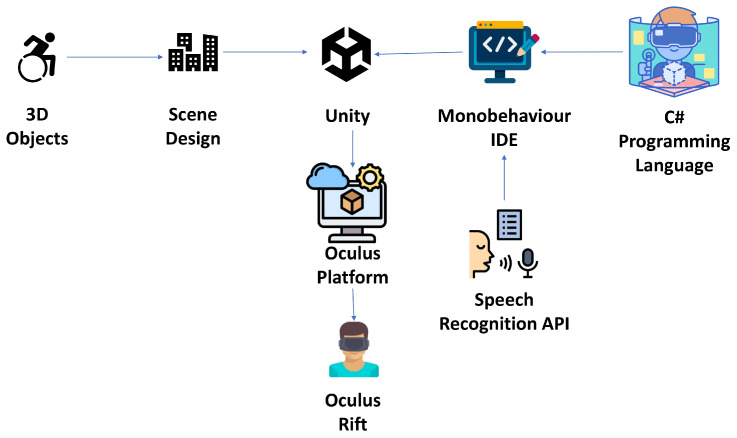
Connection diagram between Unity platform and Oculus headset.

**Figure 3 sensors-25-00530-f003:**
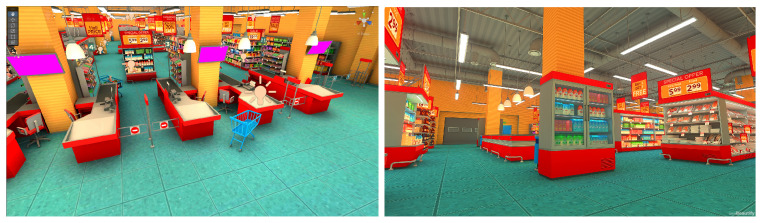
Interior view of the supermarket scene.

**Figure 4 sensors-25-00530-f004:**
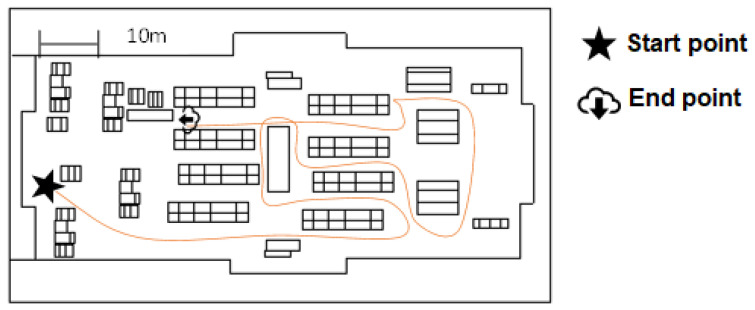
Example of a path tour in the supermarket scene. The “floor plan” illustrates the layout of the supermarket, including aisles and sections, with a line tracing the trajectory followed by a volunteer from the starting point to the end point.

**Figure 5 sensors-25-00530-f005:**
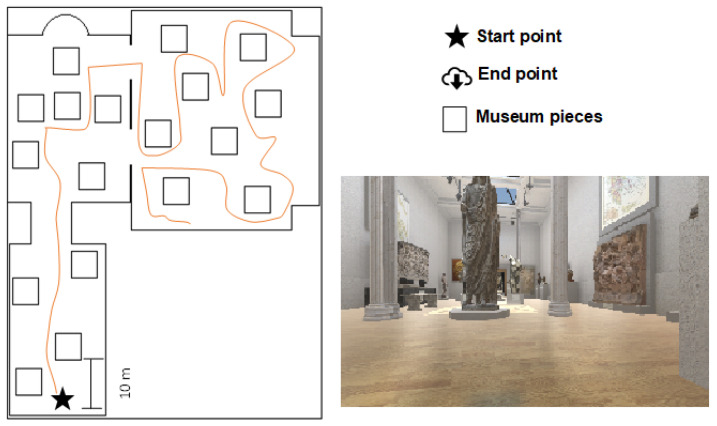
Example of a path tour in the museum scene. The “floor plan” shows the layout of the museum, including pieces and spaces, with a line tracing the trajectory followed by a volunteer from the starting point to the end point. The “interior view” provides a perspective of the museum’s environment.

**Figure 6 sensors-25-00530-f006:**
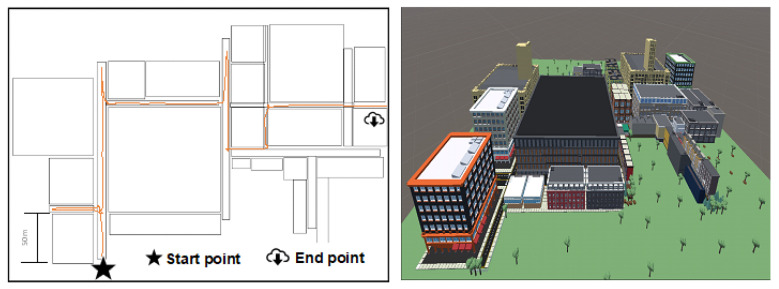
Example of a path tour on the city scene “floor plan” and “aerial view”.

**Figure 7 sensors-25-00530-f007:**
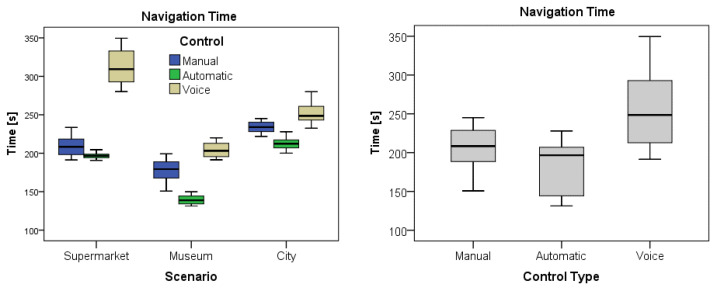
Graph of the time variable by scenario and by control.

**Figure 8 sensors-25-00530-f008:**
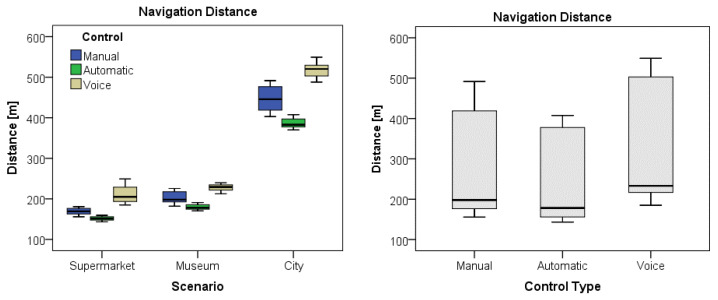
Graph of the distance variable by scenario and by control.

**Figure 9 sensors-25-00530-f009:**
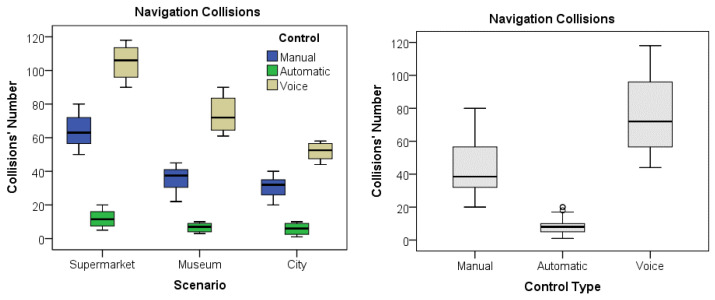
Graph of the collision variable by scenario and by control.

**Figure 10 sensors-25-00530-f010:**
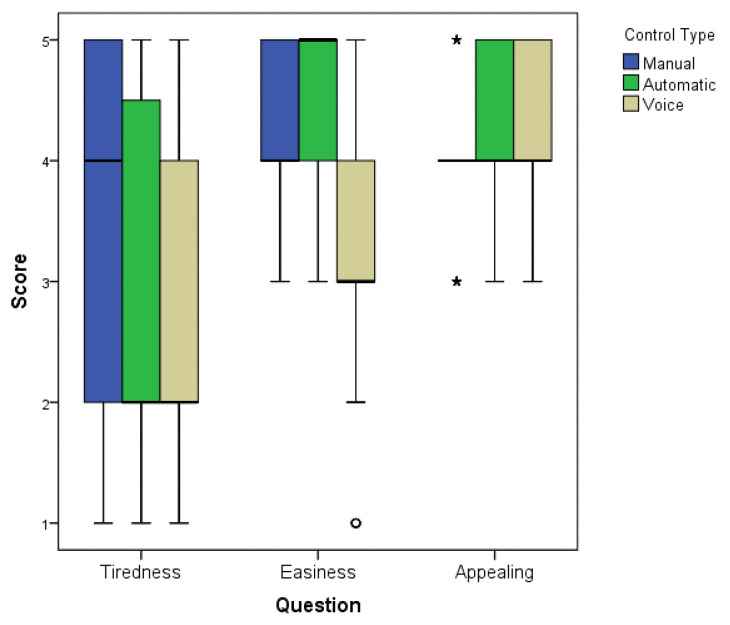
Mean difference of the statistical analysis of the perception questionnaire. The box plot shows the results for Tiredness, Ease, and Appearing across the three types of controls. Asterisks (*) indicate significant differences, and the circle highlights an outlier.

**Figure 11 sensors-25-00530-f011:**
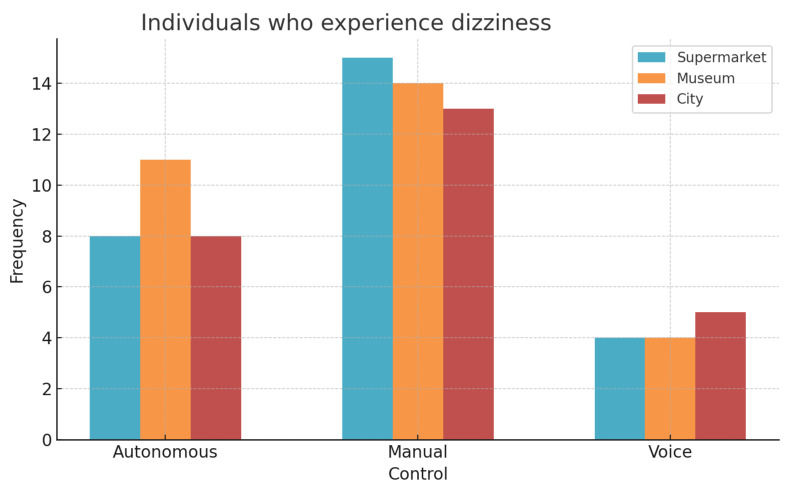
Incidence of individuals who experienced dizziness.

**Figure 12 sensors-25-00530-f012:**
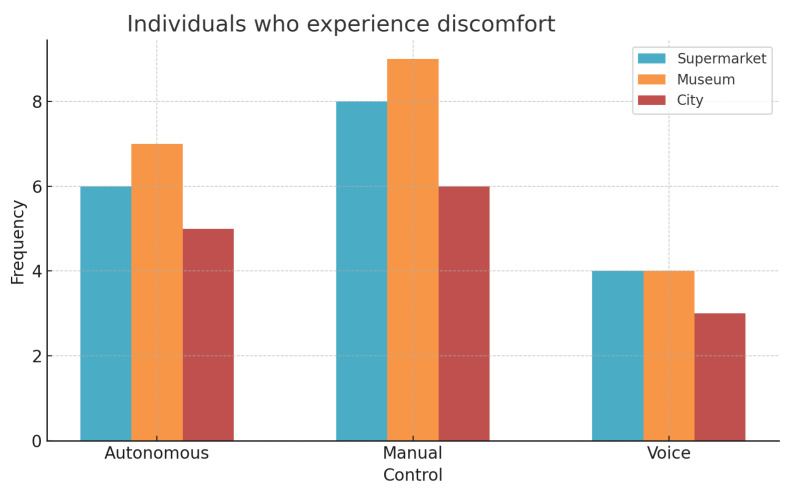
Incidence of individuals who experienced discomfort.

**Table 1 sensors-25-00530-t001:** Voice control instructions.

Instructions	
Forward	Right
Move to (forward, back, right, left)	Left
Go to (forward, back, right, left)	Turn right
Address (forward, back, right, left)	Turn left
Drive	Turn stop
Advance	Cease
Back	Discontinue
Return	Halt
Reverse (direction)	Break

**Table 2 sensors-25-00530-t002:** Likert scale for items.

Strongly Agree (7)	Often Agree (6)	Sometimes Agree (5)	Undecided (4)	Sometimes Disagree (3)	Often Disagree (2)	Strongly Disagree (1)

**Table 3 sensors-25-00530-t003:** Items and Navigation Type.

Item	Statement	Category
1	In general, I can affirm that the navigation experience was pleasant.	Navigation
2	Overall, I felt comfortable during navigation with the virtual wheelchair.	Navigation
3	I can state that I am familiar with the use of virtual reality.	Experience
4	At times, I experienced the sensation of sitting in a real wheelchair.	Realism
5	There were moments when I found navigating with the wheelchair challenging.	Control
6	During navigation, I perceived the virtual environment to be easy to comprehend.	Scenario
7	Occasionally, I experienced the sensation that interactions with objects in the scene were real.	Realism
8	At times, I felt as though I was navigating in a real environment.	Realism
9	I had the sensation that the movements of the wheelchair were driven by my own volition.	Scenario
10	Occasionally, I perceived the objects in the scene as being real.	Realism
11	In general, I found the wheelchair easy to control.	Control
12	At times, I felt as though the wheelchair was unsafe during navigation.	Control
13	Occasionally, I felt as though my body was physically seated in the virtual wheelchair.	Presence
14	I can assert that I generally felt focused during navigation.	Experience
15	There were times when I found it difficult to orient myself within the environment.	Experience
16	I had the sensation that I was effectively evading obstacles.	Experience
17	Occasionally, I experienced fatigue during navigation with the virtual wheelchair.	Experience
18	At times, I felt as though interactions with objects were occurring in parts of my real body.	Presence
19	Overall, I felt that the interaction with the virtual environment was intuitive and natural.	Presence
20	Overall, I felt as though the movements of the virtual wheelchair corresponded to my own actions.	Presence
21	In general, I have the impression that after the navigation experience, I could effectively use a real wheelchair.	Experience
22	Overall, I can assert that the navigation experience was positive.	Experience
23	Occasionally, I experienced fatigue during navigation with the virtual wheelchair.	Experience
24	At times, I experienced physical discomfort during navigation with the wheelchair.	Experience
25	Occasionally, I perceived the control of the wheelchair to be confusing.	Control

**Table 4 sensors-25-00530-t004:** Observed performance of the participants according to the scenario. * All variables showed significant differences between scenarios, achieving *p* < 10^−4^.

Scenario			
Variable	Supermarket	Museum	City
Time [s] *	239.19 ± 53.87	173.82 ± 28.75	232.66 ± 47.11
Distance [m] *	178.02 ± 28.98	203.59 ± 22.23	449.89 ± 58.21
Velocity [m/s] *	0.75 ± 0.06	1.18 ± 0.08	1.93 ± 0.11
Acceleration [m/s^2^] *	0.003 ± 0.001	0.007 ± 0.002	0.008 ± 0.0001
Collisions [count] *	60 ± 39	38 ± 28	30 ± 20

**Table 5 sensors-25-00530-t005:** Observed performance of the participants according to type of control. * Significant differences (*p* < 10^−4^) were found for all variables except Velocity [m/s], where no significant differences were observed.

Control			
Variable	Manual	Autonomous	Voice
Time [s] *	206.79 ± 25.96	182.86 ± 32.02	256.02 ± 46.95
Distance [m] *	273.03 ± 125.93	239.39 ± 105.79	319.08 ± 142.33
Velocity [m/s]	1.29 ± 0.46	1.29 ± 0.43	1.28 ± 0.58
Acceleration [m/s^2^] *	0.006 ± 0.002	0.007 ± 0.002	0.005 ± 0.003
Collisions [count] *	44 ± 17	8 ± 4	43 ± 33

**Table 6 sensors-25-00530-t006:** Descriptive statistics of the measured motion variables of the participants’ trajectories in relation to scenario and control conditions. * Significant differences (*p* < 10^−4^) were found for all variables except Distance [m], where no significant differences were observed.

Supermarket			
Variable	Manual	Autonomous	Voice
Time [s] *	209.06 ± 11.76	196.90 ± 3.82	311.62 ± 22.33
Distance [m]	169.78 ± 7.74	151.88 ± 4.99	212.4 ± 22.08
Velocity [m/s] *	0.81 ± 0.02	0.77 ± 0.01	0.68 ± 0.03
Acceleration [m/s^2^] *	0.003 ± 0.002	0.003 ± 0.006	0.002 ± 0.001
Collisions [count] *	65 ± 9.49	12 ± 4.88	104 ± 9.33
Museum			
Time [s] *	177.23 ± 14.05	139.56 ± 3.82	204.67 ± 22.33
Distance [m]	203.06 ± 7.74	179.89 ± 4.99	227.83 ± 22.08
Velocity [m/s] *	1.15 ± 0.0171	1.29 ± 0.0130	1.11 ± 0.030
Acceleration [m/s^2^] *	0.006 ± 0.007	0.009 ± 0.001	0.005 ±0.003
Collisions [count] *	36 ± 9.49	7 ± 4.873	74 ± 9.326
City			
Time [s] *	234.09 ± 11.76	212.13 ± 3.82	251.77 ± 22.33
Distance [m]	446.26 ± 7.74	386.40 ± 4.99	517.01 ± 22.08
Velocity [m/s] *	1.90 ± 0.0171	1.82 ± 0.0130	2.05 ± 0.0304
Acceleration [m/s^2^] *	0.008 ± 0.002	0.008 ± 0.006	0.008 ± 0.001
Collisions [count] *	31 ± 9.49	6 ± 4.873	52 ± 9.326

## Data Availability

For more information, please refer to the following link: https://www.diputados.gob.mx/LeyesBiblio/regley/Reg_LGS_MIS.pdf, accessed on 1 December 2024.

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
