# Peer review of "Comparison of Manual, Automatic, and Voice Control in Wheelchair Navigation Simulation in Virtual Environments: Performance Evaluation of User and Motion Sickness"

_sensors, 2025, doi:10.3390/s25020530_

Round 1

Reviewer 1 Report

Comments and Suggestions for Authors

Congrats for this topic. It's a nice work but I consider that it could be a first pilot study. Please read the file joined, I give you many considerations there

Author Response

Commented [A1]: Consider introducing other profiles (like health professionals or wheelchair users) in your research group to help you consider other perspectives.

Dear Reviewer, thank you for your comments regarding the introduction of professional health profiles. We would like to mention the following: Dr. Erik Rene Bojorges V. is a Biomedical Engineer by profession, and Dr. Miguel Ángel Padilla C. works at ICAT, an institute focused on technological and rehabilitation applications. This is a preliminary study on various controls and methods of virtually operating a wheelchair with the goal of implementing it later in controlled physical environments. In future work, we aim to involve more diverse research groups.

Commented [A2]: I don’t understand the second image. Why this posterior view?

Thank you for this observation. The original idea was to include a posterior view to demonstrate how the wheelchair navigates through one of the scenarios. However, this may cause confusion. We have replaced Image 4 with a frontal view where users can observe the environment.

Commented [A3]: That’s would be a pilotage of usage, but the problem that you don’t have account here is that you should try this procedure with real users, because they could give you an expert opinion (I agree people with visual perception problem couldn’t participate)

Thank you for your observation. We understand the importance of involving real users with wheelchair experience to obtain expert opinions. However, the primary goal of this study was to conduct an initial pilot test to assess the system's feasibility and understand the responses of participants without disabilities. This allowed us to identify areas for improvement before conducting a study with end-users.
The volunteers participating in the tests had no visual or mobility impairments, ensuring effective interaction with the visual controls and avoiding factors that might affect the initial results. In future work, we plan to expand the sample to include users with real wheelchair experience.
The “Participants” section (3.2) has been updated to emphasize the involvement of volunteers without visual or mobility issues.

Commented [A4]: Are you consider the user’s level in relationship about joystick usage or in relationship with virtual games usage. These are two important variables wich could modify the response of subjets to your intervention. If you don’t consider it’s a limitation to note.

Thank you for your observation. The users' level of experience with joysticks and their familiarity with video games are important factors that could influence their performance during the tests.
In this pilot study, we did not include an exhaustive analysis of participants' experience with joysticks or video games because the primary objective was to evaluate the system’s basic functionality and gather initial feedback on the overall interaction with the controls. However, we acknowledge that the lack of consideration of these variables is a limitation of the study.
In future work, we plan to implement a preliminary questionnaire to assess participants' experience in these areas, enabling a more detailed analysis of how these factors influence the results and improving system design and understanding.

Commented  [A5]: That’s no a real perception, you should place this subjets in a real wheel chair for knowing the real position

Thank you for your observation. Spatial perception and user experience can differ significantly when using a real wheelchair compared to a simulation. This work focused on evaluating the performance of different control systems in a virtual environment to collect initial data on interaction and system feasibility.
Using a real wheelchair during the tests was not considered at this stage due to logistical and technical limitations. However, we agree that this approach could provide a more realistic experience and allow a more accurate assessment of user interaction with the system.
In future work, we plan to conduct additional tests involving a real wheelchair in a physical environment, combined with virtual simulation, to analyze how this experience affects perception, performance, and system acceptance.

Commented [A6]: Good breefing before started?

Before starting the experiment, various instructions were provided to all participants to ensure they understood the study’s objectives and tasks and familiarize them with the controls and virtual reality glasses. During this initial session, detailed instructions on interacting with the virtual environment were given, and all questions were addressed.
Additionally, participants practiced in an empty scenario with simple objects, such as geometric shapes, to become accustomed to operating the virtual wheelchair before proceeding to the test scenarios. This ensured that all participants had a basic level of familiarity with the controls and interface before formal evaluations.
This training process was added to the “Experimental Procedure” section (3.3).

Commented [A7]: Is there a trainning process to use the controls? If not, you should consider to make an empty scenario to practice the usage before.

Thank you for your observation. A prior training process was indeed implemented to allow participants to familiarize themselves with the controls before testing in experimental scenarios. This training was conducted in an empty scenario, specifically designed with simple objects like geometric shapes (spheres, cubes, and cones), allowing users to practice basic virtual wheelchair movements such as forward, backward, turning, and stopping.
The goal of this scenario was to reduce the learning curve and ensure that participants had basic proficiency in using the controls before facing more complex tasks in the experimental scenarios. This process is detailed in the “Experimental Procedure” section (3.3).

Commented [A8]: Are you consider measured this variable as well before the intervention during the visualization of the simulation from an adequate point of view as control? Because motion sickness could be appear only with visualization

Thank you for your comment. It was not initially considered that simulation visualization alone could contribute to motion sickness symptoms. During the experiment design, we decided to measure motion sickness only after interaction with each control, assuming that this phenomenon would be more relevant during the complete experience, which includes both visualization and active control use.

Commented [A9]: This item is biased for your sample that they have not the habitude to be in a wheelchair

Thank you for your observation. The sample used in this study consists of participants without prior experience using wheelchairs, which could introduce a bias in the results, particularly regarding fatigue perception and interaction with the controls. The intent of the question was more related to a virtual wheelchair than a physical one, which might have caused confusion. The question has been revised, and the word "virtual" has been added to clarify that it refers to the virtual wheelchair.

Commented [A10]: This item is biased for your sample that they have not the habitude to be in a wheelchair

Thank you for your observation. The sample used in this study consists of participants without prior experience using wheelchairs, which could introduce a bias in the results, particularly regarding fatigue perception and interaction with the controls. The intent of the question was more related to a virtual wheelchair than a physical one, which might have caused confusion. The question has been revised, and the word "virtual" has been added to clarify that it refers to the virtual wheelchair.

Commented  [A11]: This item is biased for your sample that they have not the habitude to be in a wheelchair

Thank you for your observation. The sample used in this study consists of participants without prior experience using wheelchairs, which could introduce a bias in the results, particularly regarding fatigue perception and interaction with the controls. The intent of the question was more related to a virtual wheelchair than a physical one, which might have caused confusion. The question has been revised, and the word "virtual" has been added to clarify that it refers to the virtual wheelchair.

Commented [A12]: This item is biased for your sample that they have not the habitude to be in a wheelchair

Thank you for your observation. The sample used in this study consists of participants without prior experience using wheelchairs, which could introduce a bias in the results, particularly regarding fatigue perception and interaction with the controls. The intent of the question was more related to a virtual wheelchair than a physical one, which might have caused confusion. The question has been revised, and the word "virtual" has been added to clarify that it refers to the virtual wheelchair.

Commented [A13]: What order you use for doing each scenario? Is it always the same? Have you given a rest between controls? I ask that because the habituation effect could be involved too?

The order of the scenarios was assigned randomly to avoid any habituation effect that could influence the results. Additionally, participants were allowed to take a 5- to 10-minute rest between scenarios if they felt it necessary, helping reduce fatigue and ensure optimal performance during all tests. This observation has been added to section 4.2 of the article.

Commented [A14]: Lack of references to compare your results and to can explain them

We appreciate this observation. The discussion section has been expanded to include additional comparative references that help contextualize and explain the results of this study. These comparisons show how this study aligns with or differs from previous research and provide a stronger basis for interpreting the findings.

Commented [A15]: That isn’t correct like kind of citation

Thank you for this observation. The text was cited incompletely, and reference 61 has been corrected.

Commented [A16]: Consider to talk about security of 3 controls. It’s impressionant the better results in order to number of colissions of the autonomous system compared to the others ones.

Thank you for this comment. Based on the collected data, a detailed analysis of collision rates for each control method has been included, highlighting the superior performance of the autonomous system in minimizing collisions. This addition enhances the discussion on safety aspects and provides more information about the advantages of autonomous control during wheelchair navigation. This can be found in the discussion section.

Reviewer 2 Report

Comments and Suggestions for Authors

Please clarify the goal of the study in the abstract. It is not clear if the goal of the study is the development and usability evaluation of a virtual reality evaluation tool for wheelchair UIs or the evaluation of three wheelchair UIs or both. It is also important to mention the innovation of the proposed VRs or UIs (depending on the main goal) compared to existing technology since the research is not novel. Show p-value results and the recruited participants (able-bodied or wheelchair users) for more context.

Keywords: Use words different from your title. E.g. disabilities, training, navigation.

Introduction: Page 2, line 33-34. Testing wheelchair interfaces in simulators may not be true since the mentioned wheelchair interfaces have been developed and improved for over 20 years and tested in controlled environments. I’d suggest providing more references to support your claim or rephrasing the sentence that simulators are alternative evaluation tools for wheelchair interfaces, in addition to other benefits such as requiring less physical space and customization to the client’s daily environments.

Typically, evaluation of wheelchair interfaces has been done in controlled environments with either dummy or able-bodied participants to test their consistency, safety, and effectiveness in accomplishing the desired mobility tasks. While I agree that testing the safety of these wheelchair interfaces is important; it is difficult to ensure that the VR results will translate to the use of these interfaces in physical environments due to obvious factors such as sensor accuracy, interface interaction with existing wheelchairs, and environment conditions (e.g. weather, flooring, etc.). I’d suggest providing further justification for the benefits of evaluating wheelchair interfaces in VR compared to controlled environments.

Furthermore, the article needs to address the innovation of the proposed VR compared to research in the field. Previous research has been done by placing a wheelchair on an instrumented treadmill or double-drum test and using VR to evaluate the clients’ wheelchair skills. Lam, J. F., Gosselin, L., & Rushton, P. W. (2018). Use of virtual technology as an intervention for wheelchair skills training: a systematic review. Archives of physical medicine and rehabilitation99(11), 2313-2341.

Methods: Provide images of the manual control and Point of View of the client in the virtual environment.

How did the client interact with the autonomous control in the virtual environment? Was it used along with the voice control? Did the client select points of interest?

Voice control instructions are not clear or may seem redundant (e.g. move, Drive, and Advance sounds the same). How did you validate these instructions? Despite being a preliminary study, the delivery of instructions can affect the usability results.

Did the VR provide any haptic feedback during collisions? I’m interested in understanding the resemblance of real-world physical interaction within the virtual environment. Also, did you include background noise? This can be important when evaluating the accuracy of the voice control.

Did participants receive any training? How long did it take and what did it entail? Did they receive guidance on the location of the target objects?

Furthermore, how did time and distance measure the effectiveness of the wheelchair interfaces? These measurements may not reflect the ease of operation of the interface. For example, participants may take longer to find the target objects

Provide Units in table 6

Please elaborate on the future work and possible challenges when testing with wheelchair users and other environments.

Author Response

1) Please clarify the goal of the study in the abstract. It is not clear if the goal of the study is the development and usability evaluation of a virtual reality evaluation tool for wheelchair UIs or the evaluation of three wheelchair UIs or both. It is also important to mention the innovation of the proposed VRs or UIs (depending on the main goal) compared to existing technology since the research is not novel. Show p-value results and the recruited participants (able-bodied or wheelchair users) for more context.

Dear Reviewer, We appreciate your comments. Regarding this first point, we would like to clarify the following:

The primary goal of this study is to evaluate the usability and performance of three wheelchair user interfaces manual, automatic, and voice-controlled—using a virtual reality simulation tool.
Unlike existing physical or virtual approaches, VR tools provide a controlled and repeatable environment for prototyping and testing wheelchair UIs across various scenarios.

The abstract has been revised to highlight these points.

2) Keywords: Use words different from your title. E.g. disabilities, training, navigation.

The keywords have been revised to better reflect the research conducted.

3) Introduction: Page 2, line 33-34. Testing wheelchair interfaces in simulators may not be true since the mentioned wheelchair interfaces have been developed and improved for over 20 years and tested in controlled environments. I’d suggest providing more references to support your claim or rephrasing the sentence that simulators are alternative evaluation tools for wheelchair interfaces, in addition to other benefits such as requiring less physical space and customization to the client’s daily environments.

Thank you for your comments.
Wheelchair interfaces have indeed been developed and tested in controlled environments for over 20 years. To better reflect this fact, the introduction has been expanded to clarify that simulators, particularly virtual reality environments, serve as complementary evaluation tools rather than substitutes. Simulators offer unique advantages, such as reduced physical space requirements, customization of scenarios to replicate users' daily environments, and safe, repeatable training and testing.

Additionally, we have included more references to support the claim that simulators, including VR, are valuable tools for evaluating wheelchair interfaces and enhancing user training. These references highlight the practical benefits and therapeutic applications of VR-based simulations.

You can find these modifications in the Introduction section.

4) Typically, evaluation of wheelchair interfaces has been done in controlled environments with either dummy or able-bodied participants to test their consistency, safety, and effectiveness in accomplishing the desired mobility tasks. While I agree that testing the safety of these wheelchair interfaces is important; it is difficult to ensure that the VR results will translate to the use of these interfaces in physical environments due to obvious factors such as sensor accuracy, interface interaction with existing wheelchairs, and environment conditions (e.g. weather, flooring, etc.). I’d suggest providing further justification for the benefits of evaluating wheelchair interfaces in VR compared to controlled environments.

Dear Reviewer, Thank you for these observations.

The evaluation of wheelchair interfaces has traditionally been conducted in controlled environments with either able-bodied participants or dummies to ensure their consistency, safety, and effectiveness in accomplishing desired mobility tasks. VR simulators enable testing in a safe environment, eliminating risks associated with physical use during the design and preliminary evaluation stages.

This explanation is detailed further in Section 2, "Materials and Methods."

5) Furthermore, the article needs to address the innovation of the proposed VR compared to research in the field. Previous research has been done by placing a wheelchair on an instrumented treadmill or double-drum test and using VR to evaluate the clients’ wheelchair skills. Lam, J. F., Gosselin, L., & Rushton, P. W. (2018). Use of virtual technology as an intervention for wheelchair skills training: a systematic review. Archives of physical medicine and rehabilitation, 99(11), 2313-2341.

Thank you for these observations.
Previous research, such as the study by Lam et al. (2018), has highlighted the use of virtual reality (VR) technology for evaluating wheelchair skills, often employing platforms like instrumented treadmills or double-drum tests.

However, this study innovates by focusing not only on simulating the environment but also on integrating multiple control interfaces (manual, automatic, and voice-controlled) within an immersive virtual environment. Unlike prior studies, which have primarily concentrated on mobility simulation over controlled surfaces or using specific tools like treadmills, this approach leverages a VR simulation environment that allows users to interact more dynamically and flexibly with their surroundings, free from the constraints of physical equipment.

Furthermore, customization of the virtual environment and control interfaces enhances both the safety and effectiveness of training while providing a more accessible platform for users to practice in a simulated environment that mirrors daily usage conditions, without reliance on expensive or complex equipment.

Modifications have been made to the article, and additional references have been included to justify and highlight this research. You can find these updates in Section 2, "Materials and Methods."

6) Methods: Provide images of the manual control and point of view of the client in the virtual environment

We appreciate this observation. The original idea was to include a rear view image to show how the wheelchair moves through one of the scenarios. However, this may cause confusion. Image 4 has been replaced with a front view where users can observe the environment.

7) How did the client interact with the autonomous control in the virtual environment? Was it used along with the voice control? Did the client select points of interest?

Autonomous control was implemented as an independent option for wheelchair navigation in the virtual environment. This system allowed the wheelchair to move automatically through predefined points within the environment without direct user intervention. However, the autonomous control was not used simultaneously with voice control; both systems operated separately to evaluate the effectiveness and user preference for each control mode.
Regarding client interaction, the autonomous control enabled participants to observe the automatic movement of the wheelchair toward preconfigured points of interest in the virtual environment.

8) Voice control instructions are not clear or may seem redundant (e.g. move, Drive, and Advance sounds the same). How did you validate these instructions? Despite being a preliminary study, the delivery of instructions can affect the usability results.

Thank you for this observation.
In the initial pilot tests conducted with users, it was noted that some participants, while testing the wheelchair with voice control, occasionally forgot the specific commands needed to move to a particular point. In many cases, participants substituted the commands with similar words or stopped to ask for the correct command before proceeding.
This behavior led to the decision to implement multiple commands with similar meanings, such as "move," "drive," and "advance," providing a broader range of options. This approach allowed users greater flexibility and confidence when interacting with the system, as they could use the command they were most comfortable with or could remember more easily.
While this strategy proved effective during these tests, we acknowledge that it may affect the system's precision and clarity. Future work will aim to refine these voice control commands.
This feedback has been added to section 2.4.3 Voice Control in the manuscript.

9) Did the VR provide any haptic feedback during collisions? I’m interested in understanding the resemblance of real-world physical interaction within the virtual environment. Also, did you include background noise? This can be important when evaluating the accuracy of the voice control.

Thank you for these observations. Regarding haptic feedback during collisions, there was no haptic device to provide physical feedback (such as vibrations). Initially, we considered using a visual response during collisions, such as an on-screen message or a "shake" effect. However, during the early tests, this option was found to be somewhat distracting and annoying due to the frequency of collisions and the constant visual feedback.
As an alternative, a simple sound, similar to a "beep," was implemented for each collision. This approach allowed users to be aware of collisions without causing distractions or interfering with overall interaction.
Regarding background noise, no deliberate environmental noise was included during the tests to maintain focus on evaluating voice control. However, we understand that background noise could impact voice command accuracy and plan to consider this in future research.
These observations were included in the 2.4.2 Autonomous Control section of the manuscript. Future work will aim to incorporate haptic feedback for object interactions or collisions to enhance the sense of realism.

10) Did participants receive any training? How long did it take and what did it entail? Did they receive guidance on the location of the target objects?

Yes, participants received training before the main tests. Training was conducted in an empty space with simple objects, such as geometric shapes, to help participants familiarize themselves with the wheelchair controls in a controlled and distraction-free environment. During training, participants were guided on how to operate the wheelchair and navigate the space. The goal was to ensure they were comfortable with the control systems before starting the actual tests.
Once participants had mastered the wheelchair controls, tests were conducted randomly, without providing indications about the location of target objects. This randomization ensured a more realistic evaluation of real-world scenarios where users navigate to various points without prior knowledge of their locations.
This feedback was added to the 3.3 Experimental Procedure section of the article.

11) Furthermore, how did time and distance measure the effectiveness of the wheelchair interfaces? These measurements may not reflect the ease of operation of the interface. For example, participants may take longer to find the target objects

Time and distance may not fully reflect the ease of operation of the interfaces. In this study, these metrics were primarily used to measure overall performance and navigation efficiency. However, we agree that additional factors, such as the time spent searching for target objects, could influence the results.
To complement these metrics, a Likert-scale questionnaire was used to assess participants' subjective perceptions of ease of use, comfort, and perceived control. Additionally, aspects related to physical and psychological comfort were analyzed, including the incidence of motion sickness and discomfort during the immersive experience. These variables provide a more holistic view of the control interfaces' performance and their impact on users.
This observation has been included in the 3.6 Statistical Analysis section.

12) Provide Units in table 6

Table 6 has been updated to include the corresponding units.

13) Please elaborate on the future work and possible challenges when testing with wheelchair users and other environments.

Thank you for your comment on the need for more details regarding future work and potential challenges in testing with actual wheelchair users and in other environments. We have expanded the discussion section to include a more detailed description of anticipated challenges and key areas for future research. This new section addresses aspects such as adapting the system for users with varying abilities, integrating hardware into real-world settings, and validation in more diverse scenarios.
You can find these updates in the Conclusions section.

Round 2

Reviewer 1 Report

Comments and Suggestions for Authors

Congrats for the work. I include many considerations in the joint document that I consider that could improve the work

Author Response

1) Mark and model of head- mounted display

Dear reviewer, we appreciate your comments. Regarding the first comment, the head-mounted display device used in this study was the Oculus Quest 2, chosen for its high resolution and comfortable fit during prolonged use. This information is detailed in Section 3.3, 'Experimental Procedure,' where the equipment and setup are described. The Oculus Quest 2 was deemed suitable for this research due to its performance and the immersive experience it provides, which is crucial for evaluating wheelchair navigation in virtual environments.

2) I don't understand why you put the participants one metter away if they use an immersive device

We appreciate your comment. The reason participants were positioned one meter away from the device during the experiment, despite using an immersive device, is to optimize the overall experience and minimize potential discomfort. The distance was carefully selected based on the optimal field of view and comfort level of the participants, ensuring that the display’s immersive features were not hindered while preventing any physical strain that might occur with prolonged close proximity. Additionally, the chosen distance also ensured that the tracking sensors of the Oculus Quest 2, which operates with an external tracking system, functioned properly without interference. This setup helped maintain a balance between immersive experience and physical comfort during the experiment. This rationale is further explained in Section 3.3, "Experimental Procedure."

3) This paragraph could be better in measurements because there are no statistical consideration inside it.

We appreciate your comment. In response to your observation, we have revised and expanded section 3.6 "Statistical Analysis" to include a more detailed statistical analysis, as suggested. In the original text, we mentioned the use of time and distance as performance indicators for navigation, but we understand that these do not always directly reflect the ease of use of the interfaces.

To address this, we have specified that the time and distance data were analyzed using MANOVA to determine significant differences between the different scenarios and control types. Additionally, correlation analyses were performed to explore relationships between these metrics and questionnaire responses related to ease of use, comfort, and perceived control.

For instance, we verified whether longer navigation times correlated with lower scores on perceived ease of use. These analyses were also complemented with a Likert scale questionnaire designed to assess participants' subjective perceptions. The results from the Kruskal-Wallis tests revealed significant differences in perceived ease of use, comfort, and control across the different control types used.

4) if you put the unities in the table you don't need to include it in the legend

Thank you for your comment. We understand your point. In response, we have revised the table to include the units directly within it. Consequently, we have removed the units from the legend to avoid redundancy. We believe this update improves clarity and aligns with your suggestion.

5) These paragraphs could be better in the introduction and justification

Thank you for your suggestion. Based on your feedback, we have moved the relevant paragraphs to the introduction, as they provide important context and justification for the study. Additionally, we have added a brief introductory paragraph in the discussion section to improve the flow and enhance readability. We believe these changes make the structure more coherent and easier to follow.

6) You could compare dizziness and the habitude of the subject to use VR systems

Thank you for your suggestion. We have taken your recommendation into consideration and, in response, we have added a comparison between the level of dizziness experienced by participants and their familiarity with using virtual reality systems in the "Future Work" section (Section 7). We believe this analysis can provide deeper insight into how prior experience with virtual reality influences the physical and psychological response during wheelchair navigation. This topic will be explored further in future research.

7) And non healthy population

Thank you for your comment. In response to your suggestion, we have included the consideration of wheelchair-bound individuals in the "Future Work" section. This will allow us to explore how individuals with specific health conditions may experience wheelchair navigation differently in virtual environments. Future research will focus on evaluating the impact of health-related factors on the user experience and navigation performance.

8) I would include that it's the first step and that you will continue this research

Thank you for your comment. In response to your suggestion, we have added that this study represents the first step in a broader line of research. We emphasize in the "Future Work" section that we plan to continue this research, expanding the scope to explore additional factors and refine the results obtained in this study.

9) Most of these paragraphs don't talk about your work, so these wouldn't be in conclusions. They are very interesting to include them in the discussion and limitations sections. Consider it

In response to this comment, the conclusions section has been revised to more clearly reflect the work conducted in the study. The paragraphs that did not directly pertain to the results or conclusions were reworded to provide a clearer understanding of the work. This modification ensures that the conclusions focus on the key findings of the study, while maintaining a coherent summary of the research and its implications.

10) And they should be done in real users

Thank you for your comment. In response, we have included in the 'Future Work' section that we plan to conduct experiments with real users in future research. This will allow us to obtain more representative and applicable results to the target population, which is a crucial step in validating the findings of this study in a more realistic context.

11) For next studies done with patients, consider to pass an ethical committee and include this research in Clinical Trials

Thank you for your valuable comment. We acknowledge the importance of ethical considerations in research involving patients. We will ensure that future studies involving patients go through the necessary ethical approval processes in accordance with the standard protocols for clinical research.

Reviewer 2 Report

Comments and Suggestions for Authors

The authors addressed my comments

Author Response

Dear reviewer, we greatly appreciate your support and feedback to improve this work. We have noticed that you do not have any new comments or suggestions, for which we are truly grateful. Thank you very much for your time and valuable contribution.